# Meaningful Aging: A Relational Conceptualization, Intervention, and Its Impacts

Vivian W. Q. Lou [1,2]

1    Department of Social Work & Social Administration, The University of Hong Kong, Pok Fu Lam, Hong Kong;
     wlou@hku.hk
2    Sau Po Centre on Ageing, The University of Hong Kong, Pok Fu Lam, Hong Kong

**Abstract:** Having a meaningful life is one of the most important goals among older adults. This paper provided an overview of a programme of research and practice on meaningful aging among older Chinese adults. It firstly describes the process of developing and validating a relational conceptualization of a meaningful life (i.e., spiritual well-being) among older Chinese adults from its conceptual roots, development, and validation process since 2009 through an academic–community collaboration. In brief, a meaningful life was attributed to five relationships centered on older adults: the relationship with self, relationship with family, relationship with friends, relationship with people other than family and friends, and relationship with the environment. Secondly, the paper explains a validated assessment tool (e.g., the Spirituality Scale for Chinese Elders, (SSCE)) that was developed accordingly. Evidence-based stratified interventions derived from the conceptualization and operationalization were then introduced including a professionally led group intervention protocol, a volunteer-partner intervention protocol, and a self-help-oriented intervention, which shared eight-session core contents. Good practices in applying various interventions among older adults with diversified backgrounds (e.g., health status, age, and gender) and various service settings (e.g., community, long-term care facilities, and home visits) were then synthesized. Thirdly, feedback from stakeholders is illustrated, and good practices are discussed. In conclusion, a culturally sensitive and meaningful aging framework is timely and impactful for the globally aging world.

**Keywords:** active and spiritual well-being; Chinese; older adults; relational perspective

## 1. Introduction

In the influential book *Successful Aging*, published in the late 1980s, Rowe and Kahn stated that successful aging contains three important elements: (1) being free of disability or disease; (2) having high cognitive and physical abilities; (3) having meaningful engagements with others during interactions (Rowe and Kahn 1987, 1997). Over the past four decades, research on the risk factors for disability/disease and cognitive and physical ability has been extraordinary (see reviews in Annele et al. 2019; Cosco et al. 2013; Milte and McNaughton 2016). However, what meaningful aging means and how to effectively measure and achieve it at the individual, group, and societal levels is still an emerging area explored by scholars from multi-disciplinary backgrounds, but it remains an understudied area (de Medeiros and O'Neill 2020; Derkx et al. 2017). While being active and mentally fit are feasible and possible to be measured and articulated, what it means to have a meaningful life could be challenging to researchers, practitioners, and individuals. Meaningful aging seems to go beyond "what a person has or achieves" and "what a person did, or has been doing", but centers on "what a person is being, or being in relating with events, people, and relationships" (Taggart and Stewart-Brown 2019; Zaidi et al. 2013). In other words, meaningful aging could be better conceptualized in relation to how a person makes sense of himself and his surroundings, and the relationship between them (de Medeiros and O'Neill 2020; Derkx et al. 2017; Glanz and Neikrug 1994). One of the macro contexts that

has a significant influence on people's sense of being is cultural context. For example, in the early 1990s, Markus and Kitayama studied cultural differences on self in relation to emotion, cognition, and motivation, and they pointed out a cultural dimension of individualism and collectivism (Markus and Kitayama 1991). Later, this dimension of cultural variety was found to have impacts on social behavior and emotional regulation, well-being indicators, and interpersonal communication (Gudykunst et al. 1996; Markus and Kitayama 1994). Searching for meaning could be achieved differently in the context of diversified religious tradition, which constitutes another important context, cultural sensitivity. In some contexts, the one and only God is much more emphasized, while in others, multi-Gods are more accepted and even non-religious affiliation is reported by a significant portion of the older population, in particular in Asian countries (Skirbekk et al. 2018). Day-to-day practical experiences also suggest that a meaningful life could be different between groups with formal religious affiliation and those without, but all human beings can pursue a spiritual journey at the worldview level (Taves et al. 2018). The established literature points to an important perspective that has enlightened a series of studies on research and practice on spiritual well-being, which are synthesized in this paper. In particular, this paper (1) describes the process of developing and validating a relational conceptualization of a meaningful life (e.g., spiritual well-being) among older Chinese adults since 2009 through academic–community collaboration; (2) explains a standardized assessment tool corresponding to the relational framework and synthesizes its stratified interventions; (3) illustrates feedback from stakeholders and good practices. Details on a series of empirical studies, including sample size, undertake and retention, data collection processes, methods of analyses, outcomes, and discussions, can be found in other publications, which are not repeated here (Lou et al. 2012; Lou 2015; Tung Wah Group of Hospitals and the Sau Po Centre on Ageing, The University of Hong Kong 2020; Spiritual Well-Being Research Working Group, Tung Wah Group of Hospitals and the Sau Po Centre on Ageing, The University of Hong Kong 2013).

## 2. The Development Process of a Relational Framework of Spiritual Well-Being

### 2.1. Academic–Community Collaboration

Since 2009, a constructive academic–community collaboration has been established and developed into a long-term partnership between elderly services under one community organization and the Sau Po Centre on Ageing, The University of Hong Kong. A steering committee composed of academia and a community representative from the partner agency was established. From 2009 to 2015, while this development was in progress, meetings once every fortnight were set-up to identify research questions and discuss research methodology on sampling strategy, data collection, data analyses, piloting, and implementation. During the implementation stage, members from the partner agency extended to representatives from the community and residential and end-of-life care services. The findings were also disseminated at both the academic and community levels by both partners. Over the past few years, meetings have been held at least twice annually to review the progress of implementation, capacity building needs, and new development opportunities. During this collaboration, trustworthy relationships have been developed so that self-critique has become possible for the continuous enhancement of the work, which is illustrated below in detail.

### 2.2. Relational Worldview and Spiritual Well-Being Framework

According to a relational worldview, individuals are socialized in multiple layers of social networks, in other words, multiple interpersonal relationships that have significant impacts on how they make sense of self, time, space, and others. Interdependent self-construction is prominent in an Asian context, which argues that self-identity is attributed not only to individual-based traits, such as intelligence, honesty, and courage, but also to attributes that describe relationship features such as a person who is highly responsible to family and an obedient child to parents (Cross et al. 2003). In relating to the perception

of time in reference to the past, present, and future, cultural differences have also been identified (Fuhrman and Boroditsky 2010). Flow with time (*Shui Yu Er An*) is one of the key meta-cognitions among the Chinese that contributes to making sense of day-to-day life (Li and Cao 2021). In other words, life after life (i.e., future time) has not been emphasized as much as those from other cultural traditions with individualist worldviews. Communities are unique in their prominent collectivism cultures. In relation to space and nature, cultural tradition has played a significant role as well. Empirical studies have found that a sense of personal space and boundaries between public space and private space varies (Crawford 2021). Last but not least, people apply differentiating behavioral responses referencing three main layers of relationships: family, familiar, and stranger (Yang 1995). The literature has suggested that people from collectivist cultures are more likely to develop a relational framework, which we believe affects attitude, affection, preventive behaviors, and life satisfaction in pandemic management (Yetim 2003).

Between 2009 and 2013, a series of studies were conducted and led to a relational conceptualization of spiritual well-being for older Chinese adults (Chan et al. 2010; Lou 2015; Lou et al. 2010). Enlightened by a critical review of spirituality among the Chinese population (Shek 2010), together with practical reflections by frontline health care practitioners who took care of older adults at their end-stage of life, qualitative inquiries, including focus groups and in-depth interviews, were conducted to collect discourses on spirituality from three key stakeholders including older adults, family caregivers, and formal caregivers. During this stage, we were highly challenged by the language discourses that are culturally appropriate to describe "spirituality". Language wise, "*Ling Xin*" is used to describe spirituality and "*Lin Xin Jian Kang*" is used to refer to spiritual well-being. We found that "*Ling Xin*" was neither well comprehended by the stakeholders we interviewed nor interpreted in a consensus framework. On the contrary, stakeholder participants provided a mental representative, such as "supernatural", "ghost", and "unknown", which seemed to be associated with a sense of the uncontrollable aspects of life from the perspective of human beings. Hence, "*Sheng Ming Yi Yi*" (i.e., meaning of life) was adopted as a linguistic representative in the later stages while communicating to stakeholders regardless of the fact that in English publications spirituality and spiritual well-being were adopted. We are mindful that the series of studies synthesized in this paper were conducted with the motivation to enhance spiritual well-being in older adults using a holistic health care framework proposed by the World Health Organization, but the dimension of spiritual well-being is still a concept that is continuously under discussion (Chirico 2016). Based on a literature review, meaning in life is one of the most commonly included elements under the conceptualization of spiritual well-being, which is also in line with language discourse by lay older Chinese adults (Lou 2015).

Qualitative inquiries resulted in a multi-component conceptualization that was in line with a relational worldview in the Chinese tradition including a relationship with the self, family, friends, people other than family and friends, life and death, and nature (the environment), which was condensed by a two-stage Delphi study involving experts with a multi-disciplinary background (Lou et al. 2010). However, in scale validation studies, another great challenge was prompted—the relationship with life and death did not fit into the empirical model. We explored the literature and discussed and validated this with stakeholders again and again, and we also analyzed survey data by applying item analyses, factor analyses, and in particular through sub-group sensitivity analyses. All efforts point to only one choice in respect to the empirical findings—letting go of the element of life and death from the conceptualization. As a principal investigator leading the development of the concept for over two decades, when reflecting on this decision now, I would probably do the same if I were given another chance. Death in a time framework refers to the future, upon which the Chinese population pays less attention to as discussed in the literature (Fuhrman and Boroditsky 2010). Moreover, Chinese Confucians believe that human beings focus more on pursuing life meaning through daily practices (e.g., cultivating oneself, bringing order to the family, governing the country, and bringing peace

to all). As a Chinese Confucian quote says, "How can you know about death before you figure out the purpose of living?"

Last but not least, the challenges that were encountered involved how to define layers of relationships. When the word "family" was used under one element (e.g., relationship with family), participants were asked what the word family meant. In a cultural context that emphasizes multi-generationality and the family clan and heritage, family seems to be a highly fluid and contextualized construct that must go beyond the nuclear family structure. In addition, traditions, such as adoption (in the Chinese language "*Guo Ji*"), complicate the structure and composition of a family. During the process of developing the framework, deliberate discussions were held among research team members, followed by survey questions asking older adults about their understanding of family. This confirmed that (1) participants had a clear boundary understanding on who belongs in the family that s/he mentally refers to; (2) combinations of family members include members from extended families not only among vertical generations but also horizontal generations. In some cases, friends and adopted children were included as well. Hence, in our framework, a note was provided to empower participants to define his/her own family's mental representation. Moreover, a note was also added in the assessment tool on the spiritual scale for Chinese elders (SSCE), which is discussed below.

A scale validation study applying both exploratory and confirmatory factor analyses confirmed a five factor construct of a spiritual well-being framework (Lou 2015). The key constructs included two outcomes (i.e., the cognitive aspect of the meaning of life and affective aspect of the meaning of life), one bridging construct (i.e., transcendence), and five indicators of harmonious relationships (i.e., relationship with self; relationship with family; relationship with friends; relationship with people other than family and friends; relationship with the environment). Definitions of each construct are listed in Table 1. Moreover, a model of spiritual well-being actualization was explored through structural equation modeling. In brief, transcendence is the bridge between the five harmonious relationships and both the cognitive and affective aspects of spiritual well-being. Relationships with self, family, and friends serve as the lower layer of links between the other two relationships and transcendence (Figure 1). While the SSCE was validated by examining its reliability and structural and criteria-related validity, future studies are recommended to include measures on physiological functions, taking mortality into consideration (Lou 2015).

**Table 1.** Spiritual well-being conceptualization key constructs.

| | Component | Definition |
|---|---|---|
| Outcomes | Cognitive aspect of spiritual well-being | Perception and evaluation of the meaning of life |
| | Affective aspect of spiritual well-being | Positive and negative emotions attached to spiritual well-being |
| Bridge construct | Transcendence | Perceptions of past, present, and future events |
| Attributes of harmonious relationships | Relationship with self | Self-evaluation of the existential being of him/herself in reference to past, present, and future time |
| | Relationship with family * | Evaluation of a sense of mutual care, support, and appreciation among family members |
| | Relationship with friends | A sense of support and mutual exchange among friends |
| | Relationship with others | A sense of support and care with people other than family and friends represented in the above two constructs |
| | Relationship with environment | A sense of conformity to the surrounding environment including indoor and outdoor nature and the human environment |

Note. * Family is defined by the mental representation of each individual in a network of people that share a sense of a united togetherness (source: Lou 2015).

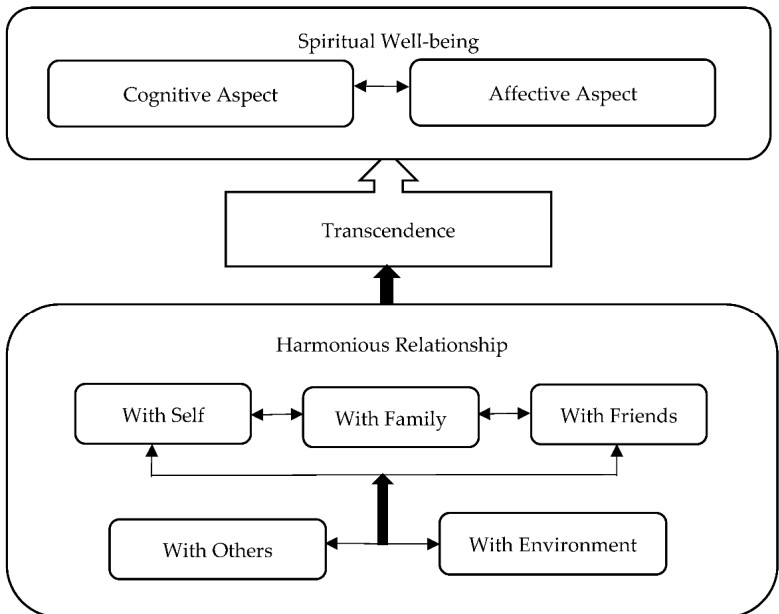

**Figure 1.** Spiritual Well-being Process Model. Source: (Lou 2015).

### 3. Assessment and Stratified Interventions with Implementation

Over the past decade, quasi-experimental studies have been designed and implmented to develop evidence-based interventions with a passion to promote spiritual well-being among the ageing population through a standardized assessment and stratification strategy, together with volunteering and tech-inclusive design (for details, please refer to Lou et al. 2012; Lou 2015; Tung Wah Group of Hospitals and the Sau Po Centre on Ageing, The University of Hong Kong 2020; Spiritual Well-Being Research Working Group, Tung Wah Group of Hospitals and the Sau Po Centre on Ageing, The University of Hong Kong 2013).

### 3.1. Standardized Assessment—Spiritual Scale for Chinese Elders

The SSCE was developed and validated through the above discussed series of studies (Lou 2015). This scale measures the above three categories of constructs: spiritual well-being, transcendence, and relationships (Table 2).

**Table 2.** SSCE sample items.

| Category of Constructs | Theme of Component | Sample Items |
| --- | --- | --- |
| Spiritual well-being 13 items | Cognitive evaluation | I believe that I have value to live in this world I am full of strength to live continuously every day |
| | Affective aspect | Have a clear conscience Disheartened |
| Transcendence 6 items | | I am proud of my life I can keep pace with societal development I found my spiritual sustenance |
| Harmonious relationships 25 items | With self | I care about my physical health I know how to take good care of myself |
| | With family | My family members show their concern for me on their own initiative I appreciate my family members |
| | With friends | My friends and I care for and support each other In times of difficulty, I have friends who can help me |
| | With people other than family and friends | I and the people around me ignore each other I have conflicts with people around me |
| | With the environment | I have a cozy living environment I maintain an orderly life |

Source: (Lou 2015).

### 3.2. Stratified Interventions

Guided by the spiritual well-being framework and spiritual process model discussed above, a risk stratification strategy was developed. As revealed in our survey studies, older adults reported varied levels of the meaning of life, as well as relationships with the self, family, friends, people other than family and friends, and the environment. Thus, one-size-fits-all interventions would not be appropriate. Hence, risk stratification strategies were adopted that would allow for interventions that target older adults with diversified needs in spiritual well-being enhancement so as to achieve holistic well-being successfully in a population (Christopher and Hemenway 2015). Two important variables are considered when we conduct stratification: the level of spiritual well-being as assessed by the SSCE and the literacy level including reading and writing and digital aspects. In the past decade, five interventions have been developed to target older adults with various needs and literacy levels. A mobile application was developed based on a self-help manual and three enhanced tech-supported functions (Spiritual Well-Being Research Working Group, Tung Wah Group of Hospitals and the Sau Po Centre on Ageing, The University of Hong Kong 2013). First, the SSCE assessment was integrated into the mobile applicaiton intervention. Second, based on the stratification scores, recommendations for prioritized intervention sessions were generated. Hence, a trained interventionist would follow up to provide session interventions accordingly. Last but not least, after completing all eight sessions, users were invited to take part in an SSCE assessment again to review changes. The mobile application has been updated from time to time and is now publicly assessable (Tung Wah Group of Hospitals and the Sau Po Centre on Ageing, The University of Hong Kong 2020). A brief summary of the five interventions is illustrated below (Appendix A).

## 4. Impacts and Good Practice Reflection

After a decade of implementation, the impacts were revealed through positive experiences and feedback from stakeholders. Good practices also reflected how and to what extent spiritual well-being could be enhanced based on both evaluation studies and feedback from participants and interventions in the various studies (Lou et al. 2012; Lou 2015; Tung Wah Group of Hospitals and the Sau Po Centre on Ageing, The University of Hong Kong 2020; Spiritual Well-Being Research Working Group, Tung Wah Group of Hospitals and the Sau Po Centre on Ageing, The University of Hong Kong 2013). In other words, the below synthesized reflections are constructed on both research-based and practice-based evidence. Finally, challenges and future directions are discussed.

### 4.1. Participant Feedback

Positive experiences were shared by the participants on how spiritual well-being was enhanced.

Breathing is a very insightful way to invite participants to get in touch with life energy moment by moment. One participant shared, " . . . breathing in happiness, breathing out negative energy . . . I look at my own smile, recall memory of time with friends, enjoy nature . . . I feel peace and joy". Another participant shared, " . . . breathing is the most insightful engagement. I will now be breathing out all my stress . . . .Let go . . . I feel peaceful after breathing out my stress, which reminds me that I deserve quality of life".

Meaningful engagement with family members was recognized and enhanced as shared by one participant, "(I had prepared a card to show my gratitude to my grandson) When I passed this card to my grandson, he felt very grateful and curious about what I had done and showed his gratitude for my efforts as well. I shared with him that I had put a lot of effort and heart into the process . . . This process enlightened me on how to enhance harmonious relationships with family members". Another participant shared, "We sometimes are shameful to express love to people we love. Now I have learned to express love in alternative ways that I could choose, for example, through sending my loved one a card".

Participating in spiritual enhancement groups expanded participants' mindsets on how to define friendship. As shared by one participant, "Now I would expect that I have the chance to meet new friends every day. I will also review photos taken during the group process to re-experience the source of life energy (life meaning)".

"Forgive at my own pace . . . ". In one of the intervention processes, forgiveness and "let go" messages were shared, with a corresponding experiential activity of using strokes on paper and/or screen to express his/her anger followed by a "wipe off". One participant shared that, "after trying to wipe off 'anger' (in the form of strokes), I felt so released in my heart. I know that I still have anger towards certain people and could not let go but I also know that I have a choice to let go when I am ready". Another participant shared, "I realized that letting go or not could be chosen by ourselves. Among our groupmates, some chose to wipe off anger but some not. I found this very insightful".

*4.2. Good Practice Reflection*

Good practices in applying various interventions among older adults with diversified backgrounds (e.g., health status, age, and gender) and various service settings (e.g., community, long-term care facilities, and home visits) could then be synthesized into one foundation and three pillars.

A working group composed of frontline social workers from the stand-alone service center mentioned above as well as professionals from residential services, community services, and academia met every three to six months since 2016 since recently to reflect on existing practices, challenges encountered, and strategies for sustained development. This multi-sector partnership serves as a solid foundation and plays a significant role in steering the three-pillar model relating to the development of knowledge transfer and dissemination for sustained impacts (Figure 2).

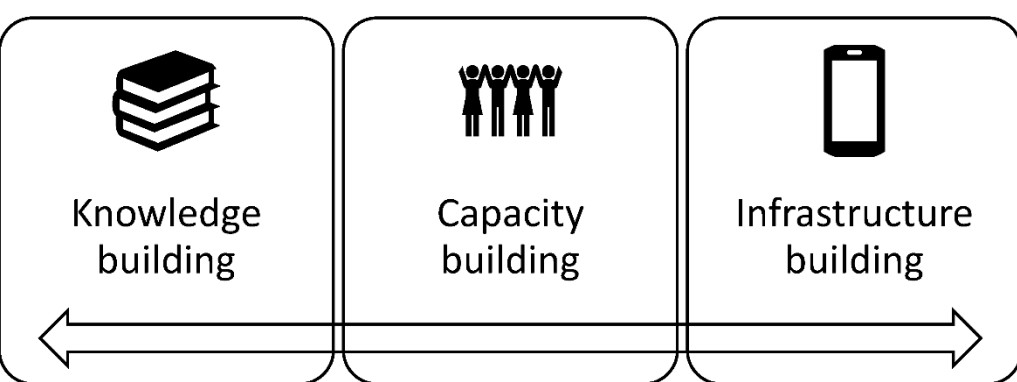

**Figure 2.** The three pillars of good practice.

The first pillar refers to continuous research activities that aim to build a knowledge base relating to evidence-based intervention protocols. As discussed above, after the first professional-led intervention protocol was established in 2012, the spiritual well-being process model has driven the development of a self-help manual, volunteer-partner protocol, medical/social collaboration train the trainer volunteer protocol, etc. All interventions are guided by the same theoretical base and apply the same standardized assessment (e.g., SSCE), which ensures a solid theoretical foundation in practice. Moreover, a core group of social work professionals work effortlessly consolidates spiritual well-being enhancement practices. This stand-alone service centre conducts one to two professional-led groups annually to older adults in community and long-term care facilities since 2016. This has been significant in achieving the sustained impact of this model within the organization and beyond.

The second pillar is capacity building targeting health, social, and spiritual care professionals. Standardized training curricula were developed based on an experiential learning framework (Matsuo 2015) (Table 3). As discussed in a previous section, spirituality

is a rarely used discourse among the Chinese population including targeted practitioners. However, holistic care (i.e., body–mind–spirit care) has often been mentioned as a service model and passion. Hence, over the years, the capacity-building courses fill in the gaps by equipping practitioners with evidence-based knowledge on spiritual well-being and practical skills to conduct assessments and corresponding interventions. Participants are enabled to start from self-reflection and peer observation on the sources of life's meaning. Participants' reflections on the sources of life energy/meaning are matched with the five sources proposed under the spiritual well-being conceptualization. Then, a mini-lecture brings abstract concepts to the participants, with research findings for illustration and support. A trial-based roleplay of the SSCE assessment serves as an active experimentation linking abstract conceptualization and reflections. Questions on what family means is the most oft-asked question, followed by the differences between spiritual well-being and psychological well-being. Further explanations of the concept of transcendence in reference to the past, present, and future can delineate concept ambiguity and clarify that spiritual well-being must embrace the concept of transcendence.

**Table 3.** Capacity building through experiential learning.

| Stages | Purpose | Activity |
| --- | --- | --- |
| Reflective observation | Increase awareness and self-consciousness on the sources of life's meaning among participants | Where does your life energy come from? What if these sources of energy fade out one by one? |
| Abstract conceptualization | Equip the concept of spirituality among Chinese elders to participants | Mini-lecture |
| Active experimentation | Facilitate participants in conducting a standardized assessment SSCE in groups of three | Role play in groups of three (interviewer, interviewee, and observer) Followed by reflection |
| Concrete experience | Practice wisdom sharing by interventionists with case illustrations | Session-by-session illustration with roleplay, reflection, and Q&A |

The last (but not least) pillar refers to infrastructure capacity. Essential infrastructure capacity building is a must-have step in order to achieve sustained impact. Over the past decade, four elements have been established: (1) A knowledge hub website with the project's details and the resources that have been built such that interested parties can attain access to them without barriers (Tung Wah Group of Hospitals and Lou 2021). (2) Capacity-building training has been offered to ensure that the relational-based spiritual well-being framework and its associated intervention protocols are introduced to interested professionals from time to time. Hence, spiritual well-being and its interventions can be shared with local practitioners' communities. Two levels of capacity building were developed including a six-hour basic-level training focused on conceptualization and the SSCE assessment and a six-hour advanced training on various intervention protocols and implementation skills (Lou 2021). (3) Tech-inclusive solutions were developed to enable self-help activities, volunteer-partner interventions, and professional-led interventions. (4) Over the years, requests have been received for the use of the SSCE in different studies, in particular by postgraduate students. We continue opening opportunities for young future scholars to further test the concept of applying the SSCE.

### 4.3. Challenges and Future Directions

Regardless of the above discussed good practices, the challenges encountered serve as another side of the same coin that motivates our passion. Regardless of the fact that the relational-based spiritual well-being theorizing has been developing for over a decade, this

concept is still not a popular discourse among the Chinese community. Great efforts are underway to share this concept to a bigger audience in the Chinese context and globally by publishing a bilingual video containing interviews of project leaders from both parties, an animation illustrating spiritual well-being and its significance in holistic health, and a new mobile application in the Apple Store that supports professionals in delivering interventions (Tung Wah Group of Hospitals 2020a, 2020b, 2020c). Secondly, since there is no designated service that is responsible for providing spiritual care and support under the current policy and service framework in Hong Kong (Social Welfare Department 2021), the practice of evidence-based interventions for enhancing spiritual well-being is nice to have but not an essential key performance indicator. When frontline workers are facing heavy workloads and uncertainties associated with social changes and service reform, there is still a long way to go in integrating this into day-to-day practices. Thirdly, reflections on capacity building and infrastructure are preliminary, and further in-depth examinations are needed to gain further support. One possible way is to conduct systematic evaluations on capacity building courses in the future. Last but not least, there are challenges as to whether this framework and its associated interventions can be tested in multi-site pilot studies and also cross over with other intervention approaches. One frontline worker with movement training tried to integrate body movement activities into the professional-led group intervention protocol and reported positive observations. A new research effort has been initiated recently and hopes to achieve another breakthrough.

## 5. Conclusions

A healthy and active aging process is a co-creation between people and his/her socio-economic and cultural context. Having a meaningful life is one of the most important purposes among older adults regardless of whether s/he has formal religious affiliation or not. While spiritual care is often regarded as being taken care of by religious organizations, in a cultural context where the majority have no formal religious affiliation, spiritual care sometimes becomes lip service without substantial content. This paper presented an overview on the historical development of a relational-oriented spiritual well-being conceptualization, assessment, and evidence-based intervention protocols developed through an academic–community partnership. The spiritual well-being process model and the SSCE serve as a solid foundation for achieving the three pillars of good practices, including knowledge capacity, manpower capacity, and infrastructure capacity. Regardless of the challenges encountered, the relational-based spiritual well-being framework was found to be well received by the Chinese community and is expected to continue to be practiced and to contribute to the development of a culturally specific conceptualization of active and healthy aging. It was concluded that by embracing a culturally sensitive conceptualization (e.g., relational-oriented conceptualization of meaningful aging/spiritual well-being), older adults, regardless of whether they have a formal religious affiliation, can embrace active aging by their own choice.

**Funding:** This piece of research received no external funding.

**Institutional Review Board Statement:** All studies summarized in the piece of summary were conducted according to Policy on Research Integrity, the University of Hong Kong.

**Data Availability Statement:** For those who are interested in data supporting the summary reported in this piece of research, please contact the author.

**Conflicts of Interest:** The author declares no conflict of interest.

## Appendix A

**Table A1.** A Summary of Evidence-Based Interventions.

| | Professional | Volunteer | Self-Help | Volunteer-Partnered | Tech-Inclusive |
|---|---|---|---|---|---|
| Theoretical base | Spiritual well-being model | | | | |
| Purpose | Enhance meaning of life of older adults | Enhance spiritual well-being among older adults with limited language literacy | Enhance understanding of sources of life meaning; sustain meaning of life | Enhance meaning of life after discharge from hospitals | Tech-enabled solution to enhance spiritual well-being |
| Target | Older adults | Older adults | Older adults | Older adults discharged from acute care | Older adults who have tech literacy |
| Inclusion criteria | - 50 or above<br>- cognitively intact<br>- can communicate via Cantonese<br>- SSCE score in lower 33rd percentile | - 50 or above<br>- cognitively intact<br>- can communicate via Cantonese<br>- SSCE score between 34th to 66th percentile | - 50 or above<br>- cognitively intact<br>- can read and write Chinese<br>- SSCE score in higher 33rd percentile | - 60 or above<br>- cognitively intact<br>- can communicate with Cantonese<br>- other criteria as specified by hospital patient resource team | - 50 or above<br>- cognitively intact<br>- digital literacy |
| Exclusion criteria | - 49 or younger<br>- cognitively impaired<br>- cannot communicate using Cantonese<br>- SSCE score not in the lower 33rd percentile | - 49 or younger<br>- cognitively impaired<br>- cannot communicate using Cantonese<br>- SSCE score not between 34th to 66th percentile | - 49 or younger<br>- cognitively impaired<br>- cannot read and write Chinese<br>- SSCE score not in the higher 33rd percentile | - 59 or younger<br>- cognitively impaired<br>- cannot communicate using Cantonese<br>- not compliant | - 49 or younger<br>- cognitively impaired<br>- digitally deprived |
| Manual | Spirituality vitalizer: Intervention manual for professionals (Lou et al. 2012) | Fu Le Man Xin (Spiritual Well-Being Research Working Group, Tung Wah Group of Hospitals and the Sau Po Centre on Ageing, The University of Hong Kong 2013) | Fu Le Man Xin (Spiritual Well-Being Research Working Group, Tung Wah Group of Hospitals and the Sau Po Centre on Ageing, The University of Hong Kong 2013) | Fu Le Man Xin (Lou and Dai 2015) | Fu Le Man Xin mobile App (Tung Wah Group of Hospitals and the Sau Po Centre on Ageing, The University of Hong Kong 2020) |
| Contents | 8 sessions<br>- Open our heart<br>- Breathe our life<br>- Be good to self<br>- Power of family<br>- Friendship<br>- Interpersonal communication<br>- Live with nature<br>- Embrace blessing | Usually 4 sessions<br>- Open our heart and breathe our life<br>- Power of family and friendship<br>- Interpersonal community and live with nature<br>- Embrace blessing | 8 sessions<br>- Open our heart<br>- Breathe our life<br>- Be good to self<br>- Power of family<br>- Friendship<br>- Interpersonal communication<br>- Live with nature<br>- Embrace blessing | 6–8 sessions according to recommendations after baseline SSCE score | 4–6 sessions according to recommendations after baseline SSCE assessment |

**Table A1.** *Cont.*

| | Professional | Volunteer | Self-Help | Volunteer-Partnered | Tech-Inclusive |
|---|---|---|---|---|---|
| Session plan | Standardized session plan with manual<br>Except for the first session, each session is achieved in three main sub-sessions:<br>-review life meaning sources and breathing exercise<br>-main theme activity<br>-homework assignment and round up | Self-help book with integrated sessions of 2 and 3, and 4 and 5 | The self-help book designed standardized session flow and exercises for readers to decide | The self-help book designed standardized session flow and exercises for readers to decide | The mobile application sessions were developed based on self-help book standardized session flow and exercises for readers to decide |
| Interventionist competence | Professionals with 12 hours training | Trained volunteers with 12 hours training | At least junior secondary school education; visual capacity to read the self-help manual | 8 hours standardized training to equip volunteers with competence to comprehend the spiritual well-being model and also deliver intervention sessions guided by the self-help booklet | Self-initiated can be achieved without specific training; volunteers can be trained to use the mobile application to deliver intervention to older adults with limited and/or no digital literacy |
| Evaluation | Pre and post evaluation using SSCE | Pre and post evaluation | Reflective notes are encouraged for readers | Pre and post assessment using SSCE | SSCE assessment before and after intervention has been integrated into the mobile application |

Sources: (Lou 2015; Lou and Dai 2015; Lou et al. 2012; Spiritual Well-Being Research Working Group, Tung Wah Group of Hospitals and the Sau Po Centre on Ageing, The University of Hong Kong 2013; Tung Wah Group of Hospitals and the Sau Po Centre on Ageing, The University of Hong Kong 2020). Note: Under specific intervention studies, participants were referred by social workers who were introduced to the study objective and inclusion and exclusion criteria. Trained interventionist would then invite participants for a brief meeting to assess inclusion and exclusion criteria and invite those who fulfilled. Participation was voluntary with an uptake rate of around ninety percent since all participants were recruited by social workers who had relationships with potential participants. Retention rate was around eighty percent, mainly due to death of participants, clinical decline in health status, severe communication difficulties due to hearing impairment and/or other clinical conditions, and change of family conditions.

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
