# Peer review of "Meaningful Aging: A Relational Conceptualization, Intervention, and Its Impacts"

_socsci, doi:10.3390/socsci11010010_

Round 1

Reviewer 1 Report

This paper addresses a very interesting topic.

Yet, I don't understand the aim of the paper.

There are no data - even if the paper is structured as a research paper. There is a discussion that refers to a validation of a scale presented in another study, and a theoretical discussion of a risk stratification strategy, and the authors comment on participants' experience, without providing any data.

Either the authors reframe and rewrite the paper as a commentary/ opinion paper- or they add a clear methodology, relevant data, and statistical analysis for this to be considered a research paper.

A third option would be writing a systematic review or even a meta-analysis on the interventions reported in Appendix 1.

In the current form, the paper looks like a hybrid, and the aim of the contribution remains unclear.

Author Response

Reviewer 1

#1This paper addresses a very interesting topic.

Response: Appreciate the comment.

#2Yet, I don't understand the aim of the paper.

Response: Thanks for the comment. The purpose of the paper was stated in the revised version in abstract and on page 2.

In particular, this paper (1) describes the process of developing and validating a relational conceptualization of a meaningful life (e.g., spiritual well-being) among older Chinese adults since 2009 through academic–community collaboration; (2) explains a standardized assessment tool corresponding to the relational framework and synthesizes its stratified interventions; and (3) illustrates feedback from stakeholders and good practices. Details on a series of empirical studies, including sample sizes, undertake and retention, data collection processes, methods of analyses, outcomes, and discussions, can be found in other publications, which are not repeated here (Lou, et al., 2012; Lou, 2015b; Tung Wah Group of Hospitals and the Sau Po Centre on Ageing, The University of Hong Kong, 2020; Spiritual Well-Being Research Working Group, Tung Wah Group of Hospitals and the Sau Po Centre on Ageing, The University of Hong Kong, 2013).”     

#3There are no data - even if the paper is structured as a research paper. There is a discussion that refers to a validation of a scale presented in another study, and a theoretical discussion of a risk stratification strategy, and the authors comment on participants' experience, without providing any data.

Either the authors reframe and rewrite the paper as a commentary/ opinion paper- or they add a clear methodology, relevant data, and statistical analysis for this to be considered a research paper.

A third option would be writing a systematic review or even a meta-analysis on the interventions reported in Appendix 1.

In the current form, the paper looks like a hybrid, and the aim of the contribution remains unclear.

Response: Thanks for the comments. The revised version positions the paper as a commentary/opinion paper by 1) revising sub-headings to highlight the three objectives; and 2) stating it clearly in abstract and conclusion session (page 1, 8-9). 

Reviewer 2 Report

Thank you for the opportunity to review this interesting article about relational conceptualization and meaningful ageing. I commend the authors for their work in this important area, and on their projects, interventions, and development of concepts.

Overall: this paper attempts to cover a lot of ground from concept to development of an assessment tool, application, and impact. In doing so, it becomes difficult, in places, for the reader to understand the aim of the article and follow a logical pathway through the detail included.

Perhaps some re-structuring of the article may be helpful to first guide the reader along a clear pathway of tool and concept development from original construct to final application, before exploring reflections/feedback from both participants and professionals involved. If so, perhaps a simple figure/timeline may be helpful to guide the reader through the author’s journey and research developments? Or some clearer sub-headings within the text?

It may also benefit from a succinct overview of important detail about the process, especially as regards the validation of the Spiritual Scale for Chinese Elders (SSCE), and including sample sizes, uptake and retention, data gathering, methods of analysis, and clarity around any agreed outcome measures and impact. Also, an insight into current levels of uptake and use, and in which contexts i.e., further academic research, care, or nursing facilities.  I do appreciate that a more comprehensive level of detail may be in the papers referenced (and in those already published by the authors), but in this instance, a summary may assist the reader of this article to better understand the links between the development and implementation phases, and subsequent impact.

The use of technology via an App-based solution is timely and fascinating. This was highlighted in the abstract but sadly lacked much further detail in the main results and discussion sections about access, uptake, or impact. It was unclear if this was currently active or still in planning phase, and if so how and when it would become available.

Introduction

  • Consider referencing some seminal papers or research on ‘healthy ageing’ i.e., it is stated that this area of research has been extraordinary, and it may be helpful to clarify and support with evidence of developments in this field
  • It may be useful to clarify how the researchers define being mentally fit and active as measurable concepts, perhaps with references
  • Have other researchers attempted to measure ‘meaningful ageing’? or attempted to define this?
  • Academic-community collaboration: this sounds fascinating but lacks detail for the reader i.e., how were these collaborative relationships established, how many organisations are involved, how regular are the meetings, and how and when were the findings disseminated? Are these collaborations still active?
  • Lines 113-115: This section may benefit from more detail around the ‘various effects’ referenced to explain what these were, when they were conducted and by whom. Also ‘analysed data in various ways’ would benefit from more detail
  • Line 117: I am unsure at to whom the ‘I’ refers in this statement: is this a reflection from the author?
  • Discussions around ‘family’ are fascinating: it is possible to clarify at which point of the development of the framework that this shift was adopted?

Materials and Methods

  • SSCE: I am not clear when was this validated, and for whom i.e., elders may be defined by different chronological ages in different cultures. Was it in the Lou (2015b) paper?
  • If may be helpful for the reader if figures and tables could be explicitly linked to published papers i.e., is Table 2 based on the validated/published SSCE paper?
  • Appendix 1: would it be possible to offer the reader more detail around uptake and retention of participants using these different interventions? This may fit well with and support the positive qualitative feedback reported (line 198).

Discussion

  • Experiences by participants: insightful and colourful quotes from participants! It may be useful for the reader to know where and how this feedback was shared and in which of the intervention the participant had taken part.
  • Good practice reflection: good explanation of the group diversity. May also be useful to have clarity around how often is ‘meeting regularly’. Has this working group also been running for nearly a decade alongside the authors research? Some qualitative feedback (quotes) from professionals involved (if available) may be helpful here to expand upon perceived challenges and solutions and to detail the opinions of those involved in development and delivery.
  • Line 252: The stand-alone service centre and professionals involved in this sounds great! It may be helpful for the reader to understand how they are supporting the model i.e., how many practitioners and elders do they support? What are the levels of uptake and change year on year?

Challenges and future directions

  • Line 298: is it possible to give more detail on what these efforts are?
  • I admire the authors passion and enthusiasm for their new research efforts, and support of meaningful ageing for older adults

Author Response

Reviewer 2

Thank you for the opportunity to review this interesting article about relational conceptualization and meaningful ageing. I commend the authors for their work in this important area, and on their projects, interventions, and development of concepts.

 #1overall: this paper attempts to cover a lot of ground from concept to development of an assessment tool, application, and impact. In doing so, it becomes difficult, in places, for the reader to understand the aim of the article and follow a logical pathway through the detail included.

Perhaps some re-structuring of the article may be helpful to first guide the reader along a clear pathway of tool and concept development from original construct to final application, before exploring reflections/feedback from both participants and professionals involved. If so, perhaps a simple figure/timeline may be helpful to guide the reader through the author’s journey and research developments? Or some clearer sub-headings within the text?

It may also benefit from a succinct overview of important detail about the process, especially as regards the validation of the Spiritual Scale for Chinese Elders (SSCE), and including sample sizes, uptake and retention, data gathering, methods of analysis, and clarity around any agreed outcome measures and impact. Also, an insight into current levels of uptake and use, and in which contexts i.e., further academic research, care, or nursing facilities.  I do appreciate that a more comprehensive level of detail may be in the papers referenced (and in those already published by the authors), but in this instance, a summary may assist the reader of this article to better understand the links between the development and implementation phases, and subsequent impact.

The use of technology via an App-based solution is timely and fascinating. This was highlighted in the abstract but sadly lacked much further detail in the main results and discussion sections about access, uptake, or impact. It was unclear if this was currently active or still in planning phase, and if so how and when it would become available.

Response: Thanks for comments. The revised version has formulated as a commentary/opinion paper that aims to achieve three objectives (stated in abstract on page 2, and introduction on page 8-9). On empirical study details, detailed references were provided for readers who are interested on page 2 (“Details of serious of empirical studies, including sample sizes, undertake and retention, data collection process, methods of analyses, outcomes and discussions could be found in other publications, which would not be repeated here (Lou, et al., 2012; Lou, 2015b; Tung Wah Group of Hospitals & Sau Po Centre on Ageing The University of Hong Kong, 2020; Spiritual Well-being Research Working Grouop Tung Wah Group of Hospitals, Sau Po Centre on Ageing The University of Hong Kong, 2013).”)  The mobile app application has been developed and now is in the market. The revised version added a brief on this with a reference for further reading on page 6.   

#2Introduction

 Consider referencing some seminal papers or research on ‘healthy ageing’ i.e., it is stated that this area of research has been extraordinary, and it may be helpful to clarify and support with evidence of developments in this field

  • It may be useful to clarify how the researchers define being mentally fit and active as measurable concepts, perhaps with references
  • Have other researchers attempted to measure ‘meaningful ageing’? or attempted to define this?
  • Academic-community collaboration: this sounds fascinating but lacks detail for the reader i.e., how were these collaborative relationships established, how many organisations are involved, how regular are the meetings, and how and when were the findings disseminated? Are these collaborations still active?
  • Lines 113-115: This section may benefit from more detail around the ‘various effects’ referenced to explain what these were, when they were conducted and by whom. Also ‘analysed data in various ways’ would benefit from more detail
  • Line 117: I am unsure at to whom the ‘I’ refers in this statement: is this a reflection from the author?
  • Discussions around ‘family’ are fascinating: it is possible to clarify at which point of the development of the framework that this shift was adopted?

Response: Thanks for comments. Literature on concept discussion and measurement on meaningful aging was added in the revised version (page 1). Academic-community collaboration session was further elaborated (page 2). In regard with Lines 113-115, revised manuscript clarified what does it mean various ways in efforts and data analyses (page 3). In regard with Line 117, “I” refer to the author, which is stated clearly in the revised version (page 3). In regard with “family”, revised manuscript clarified the concern on at which point of the development on page 4.

#3Materials and Methods

  • SSCE: I am not clear when was this validated, and for whom i.e., elders may be defined by different chronological ages in different cultures. Was it in the Lou (2015b) paper?
  • If may be helpful for the reader if figures and tables could be explicitly linked to published papers i.e., is Table 2 based on the validated/published SSCE paper?
  • Appendix 1: would it be possible to offer the reader more detail around uptake and retention of participants using these different interventions? This may fit well with and support the positive qualitative feedback reported (line 198).

Response: Thanks for comments. The validation of SSCE was detailed in Lou, V. W. (2015b). Spiritual well-being of Chinese older adults: conceptualization, measurement and intervention. Springer. The revised manuscript added the citation on page 5. Sources for Table 1, 2, Figure 1, and Appendix I were added in the revised manuscript on page 4-6, 14. A note was added under Appendix I providing more detail around update and retention of partici8pants using different interventions (page 13)

#4Discussion

 Experiences by participants: insightful and colourful quotes from participants! It may be useful for the reader to know where and how this feedback was shared and in which of the intervention the participant had taken part.

  • Good practice reflection: good explanation of the group diversity. May also be useful to have clarity around how often is ‘meeting regularly’. Has this working group also been running for nearly a decade alongside the authors research? Some qualitative feedback (quotes) from professionals involved (if available) may be helpful here to expand upon perceived challenges and solutions and to detail the opinions of those involved in development and delivery.
  • Line 252: The stand-alone service centre and professionals involved in this sounds great! It may be helpful for the reader to understand how they are supporting the model i.e., how many practitioners and elders do they support? What are the levels of uptake and change year on year?

Response: Thanks for comments. Experiences by participations were quoted from varies studies and stated in the revised manuscript (page 7). Details on good practice reflections were added on page 7. Qualitative feedbacks from professionals involved have been shared during regular meetings. However, since these meetings are working group meetings discussing promotion and development of varies projects, informed consents have not been obtained. Hence, quotes could not be shared in the manuscript. On the stand-alone service centre, trained practitioners conduct 1-2 groups professional led groups to older adults in community and long-term care facilities since 2016 (page 8).

#5 Challenges and future directions

 Line 298: is it possible to give more detail on what these efforts are?

  • I admire the authors passion and enthusiasm for their new research efforts, and support of meaningful ageing for older adults

Response: Thanks for comments and really appreciate. Efforts in promoting were illustrated in the revised manuscript (page 9).

Reviewer 3 Report

The paper aimed to describe the work of the authors on meaningful life among older Chinese adults. It presented the conceptualization, development and validation of the assessment tools for the construct of spirituality. This work is important as it advances the research on the identification and understanding of culturally unique features of successful, healthy and active aging in Chinese populations. However, although the paper did a good job in describing the social work practices that the authors developed and implemented over the years, descriptions of the research that underlies the practices were inadequate or lacking in details, thus underscoring the need for clarifications in the writing and the need for more rigorous studies. 

Additionally, building on the research, the paper described practices (e.g. interventions) that aimed to enhance spiritual well-being. The authors claimed that the practices are effective; however, the evidence they presented were anecdotal experiences and testimonials (p. 6, section 3.1) and therefore scientifically weak according to the hierarchy of evidence. This highlights that systematic investigations of the effectiveness of the practices (e.g. by using a RCT) are in order.

Overall, the paper presented promising ideas and models developed from valuable, long-standing community-academic partnership. However, again, the absence of strong evidence underscores the need for more rigorous research to test and refine the proposed ideas and models, and to evaluate if the interventions developed can be considered evidence-based practice. This is a major weakness of the work as presented. The authors are encouraged to address this as a limitation and a direction for future research in the Discussion (p. 8, Section 3.3). Below I present questions and comments for the authors to address in order to improve the paper. 

Specific substantive comments about the paper are as follows:

  • The link between a meaningful life and spiritual well-being needs better articulation. For example, did the authors intend to say that having spiritual well-being is the same as having a meaningful life (i.e., the two terms are interchangeable), that spiritual well-being is a defining feature of a meaningful life, or that spiritual well-being contributes to a meaningful life? These 3 kinds of possible conceptual links are distinct from one another. The authors should clarify their perspective.   
  • Measurement of most constructs in the spiritual well-being framework (p.151, line 139) relied on subjective self-report. Were the constructs highly correlated, suggesting either conceptual overlap or the need for better measures to differentiate the constructs? Were there any objective indicators for the constructs? Were the constructs predictive of objective outcomes such as physiological functioning, morbidities and mortality? It would be informative to address these questions in the paper.
  • It is unclear how a process model of spiritual well-being actualization can be “achieved through structural equation modeling” (p. 151, line 146). For instance, did the authors use SEM to examine longitudinal data that allow for assessment of change over time and inference of causality that led to the proposed model? Please correct or elaborate.
  • The “one foundation and three pillars” of good practice in the Discussion section (p. 6, section 3.2) served as a good overview of the overall research and practice plan. Can the authors describe the available evidence on the effectiveness of the various resources (e.g. manuals, protocol, curricula) that they developed over the years to support their claim that the resources were evidence-based as opposed to, or in addition to, practice-based?
  • It appears that considerable capacity and infrastructure building had taken place before a solid knowledge base was established (pp. 7-8). If that was indeed the case, the authors need to discuss the absence or dearth of empirical evidence for the capacity and infrastructure being built as a limitation. On the other hand, if that was not the case, the authors need to present stronger research evidence, with citations, to convince the readers that the capacity and infrastructure building were data driven.

Comments about technical issues are as follows:

  • More precise use of technical terms is needed throughout the paper. For instance, to develop evidence-based interventions, did the authors “designed and implemented” “experiments” (p. 5, line 17) which involve the testing of hypotheses in controlled conditions to examine causal relations between independent and dependent variables, or did they conduct correlational studies in naturalistic settings?
  • The whole paper would benefit from professional editing so that it can become more readable. Also, currently the paper included mixed use of writing in the subjective first-person plural stance (e.g. p.2, line 97, “we were highly challenged…”), the subjective first-person singular stance (e.g., p. 3, line 117, “When I reflect on this decision now, I would….”), and the objective third person passive voice (e.g. p. 3, line 131, “deliberate discussions were had among research team members”).  The authors should adopt a consistent style of writing throughout, following the preferred style of this Journal.
  • Figures
    • Figure 1: One of the 5 boxes in the bottom only has the word “With” in it. Please fix that and make sure that all presented information is complete and clear. 

Author Response

Reviewer 3

#1The paper aimed to describe the work of the authors on meaningful life among older Chinese adults. It presented the conceptualization, development and validation of the assessment tools for the construct of spirituality. This work is important as it advances the research on the identification and understanding of culturally unique features of successful, healthy and active aging in Chinese populations. However, although the paper did a good job in describing the social work practices that the authors developed and implemented over the years, descriptions of the research that underlies the practices were inadequate or lacking in details, thus underscoring the need for clarifications in the writing and the need for more rigorous studies. 

Additionally, building on the research, the paper described practices (e.g. interventions) that aimed to enhance spiritual well-being. The authors claimed that the practices are effective; however, the evidence they presented were anecdotal experiences and testimonials (p. 6, section 3.1) and therefore scientifically weak according to the hierarchy of evidence. This highlights that systematic investigations of the effectiveness of the practices (e.g. by using a RCT) are in order.

Overall, the paper presented promising ideas and models developed from valuable, long-standing community-academic partnership. However, again, the absence of strong evidence underscores the need for more rigorous research to test and refine the proposed ideas and models, and to evaluate if the interventions developed can be considered evidence-based practice. This is a major weakness of the work as presented. The authors are encouraged to address this as a limitation and a direction for future research in the Discussion (p. 8, Section 3.3). Below I present questions and comments for the authors to address in order to improve the paper. 

Response: Thanks for comments. The revised manuscript is shaped into a commentary/opinion paper with changed sub-headings. Details of serious of empirical studies, including sample sizes, undertake and retention, data collection process, methods of analyses, outcomes and discussions could be found in other publications, which would not be repeated here (Lou, et al., 2012; Lou, 2015b; Tung Wah Group of Hospitals & Sau Po Centre on Ageing The University of Hong Kong, 2020; Spiritual Well-being Research Working Group Tung Wah Group of Hospitals, Sau Po Centre on Ageing The University of Hong Kong, 2013). Limitations of the paper was revised acknowledging needs for future study.

Specific substantive comments about the paper are as follows:

  • #2 The link between a meaningful life and spiritual well-being needs better articulation. For example, did the authors intend to say that having spiritual well-being is the same as having a meaningful life (i.e., the two terms are interchangeable), that spiritual well-being is a defining feature of a meaningful life, or that spiritual well-being contributes to a meaningful life? These 3 kinds of possible conceptual links are distinct from one another. The authors should clarify their perspective.   

Response: Thanks for comments. Revised manuscript added a brief discussion to state the positioning of the paper on page 3.

  • #3Measurement of most constructs in the spiritual well-being framework (p.151, line 139) relied on subjective self-report. Were the constructs highly correlated, suggesting either conceptual overlap or the need for better measures to differentiate the constructs? Were there any objective indicators for the constructs? Were the constructs predictive of objective outcomes such as physiological functioning, morbidities and mortality? It would be informative to address these questions in the paper.

Response: Thanks for comments. The SSCE was developed as an attitude test asking participants to evaluate their perceptions on eight dimensions of spiritual well-being. Reliability and validity of SSCE was examined by psychometrics including factor, structural validity and criteria-related validity by association with positive and negative affect, purpose in life, and quality of life (Lou, 2015b). The revised manuscript added a limitation on not including objective testing into the validation study, which deserve future study (page 4).

#4It is unclear how a process model of spiritual well-being actualization can be “achieved through structural equation modeling” (p. 151, line 146). For instance, did the authors use SEM to examine longitudinal data that allow for assessment of change over time and inference of causality that led to the proposed model? Please correct or elaborate.

Response: Thanks for comments. The revised manuscript correct it by stating ”a model of spiritual well-being actualization has been explored through structural equation modelling” (page 4).

  • #4The “one foundation and three pillars” of good practice in the Discussion section (p. 6, section 3.2) served as a good overview of the overall research and practice plan. Can the authors describe the available evidence on the effectiveness of the various resources (e.g. manuals, protocol, curricula) that they developed over the years to support their claim that the resources were evidence-based as opposed to, or in addition to, practice-based?

Response: Thanks for the comment. Revised manuscript acknowledged that reflections on good practices are based on both research-based and practice-based evidences (page 6).

  • #5It appears that considerable capacity and infrastructure building had taken place before a solid knowledge base was established (pp. 7-8). If that was indeed the case, the authors need to discuss the absence or dearth of empirical evidence for the capacity and infrastructure being built as a limitation. On the other hand, if that was not the case, the authors need to present stronger research evidence, with citations, to convince the readers that the capacity and infrastructure building were data driven.

Response: Thanks for comments. Revised manuscript acknowledged that capacity building and infrastructure reflections are preliminary that deserve further examination (page 9).

#6Comments about technical issues are as follows:

  • More precise use of technical terms is needed throughout the paper. For instance, to develop evidence-based interventions, did the authors “designed and implemented” “experiments” (p. 5, line 17) which involve the testing of hypotheses in controlled conditions to examine causal relations between independent and dependent variables, or did they conduct correlational studies in naturalistic settings?
  • The whole paper would benefit from professional editing so that it can become more readable. Also, currently the paper included mixed use of writing in the subjective first-person plural stance (e.g. p.2, line 97, “we were highly challenged…”), the subjective first-person singular stance (e.g., p. 3, line 117, “When I reflect on this decision now, I would….”), and the objective third person passive voice (e.g. p. 3, line 131, “deliberate discussions were had among research team members”).  The authors should adopt a consistent style of writing throughout, following the preferred style of this Journal.
  • Figures
    • Figure 1: One of the 5 boxes in the bottom only has the word “With” in it. Please fix that and make sure that all presented information is complete and clear. 

Response: Thanks for comments. Revised manuscript revised technical terms with references (page 5). Presentation on Figures has been enhanced. The whole paper was further edited by professional services to improve readability (a certificate was attached).

Round 2

Reviewer 1 Report

The authors did an excellent job in restructuring the paper as an opinion paper. Now the contribution has a clear "identity" and the information is structured and presented in a much more clear way.

Author Response

Thanks for the reviewer's affirmation.  

Reviewer 2 Report

Appreciate all your time and attention to detail by addressing my comments. Thank you. I very much enjoyed reading your revised paper.

Author Response

Thanks for the reviewer's affirmation

Reviewer 3 Report

The authors have addressed most of my previous comments satisfactorily. Based on this revision, here are some minor comments to further improve the paper:

 1) Rather than calling this paper an "opinion paper" or a "commentary", I think it would be more appropriate to call it an "overview" of a programme of research and practice on meaningful aging among older Chinese adults. If the authors agrees with this perspective, please change the terminology throughout the paper.

2) Some grammatical errors remain in the paper. Please double-check and fix them. For example:

  • p.2, line 65: "... sample size, undertake and retention, ... "
  • p.8, line 305: "This stand alone service centre conduct one to two professional led group annually..."
  • p.9, Table 3, in the row for "Active experimentation" and the column for "Activity": "Roleplay..." should be "Role play..."
  • p. 14, line 453: "conditiond" should be "conditions"

3) Better word choices would contribute to clearer communication of the authors' ideas. For example:

-- p.1, Abstract, line 9: Suggest to replace "most important purposes" with "most important goals". With the replacement, this programme of work can be linked to the literature on goals and goal pursuit in social psychology. This suggestion may have implications for the underpinning science-- it is up to the authors to decide whether to follow this suggestion.

-- P.9, line 350: Suggest to replace "wider readers" with "a bigger audience".

-- P.9, line 364: Suggest to replace "can be evidenced in multi-site pilots..." with "can be tested in multi-site pilot studies..."

-- p.10, line 381-383: Suggest to replace "... and will continue to be practiced with a significant contribution towards enhancing a culturally specific explanation of active and healthy aging globally" with "... and is expected to continue to be practiced and to contribute to the development of a culturally specific conceptualization of active and healthy aging".  It would be prudent to avoid making definite statements when the evidence from rigorous scientific research is not yet available. 

Author Response

The authors have addressed most of my previous comments satisfactorily. Based on this revision, here are some minor comments to further improve the paper:

 #1) Rather than calling this paper an "opinion paper" or a "commentary", I think it would be more appropriate to call it an "overview" of a programme of research and practice on meaningful aging among older Chinese adults. If the authors agrees with this perspective, please change the terminology throughout the paper.

Response: Thanks for the suggestion and the terminology has been changed throughout the paper.

#2) Some grammatical errors remain in the paper. Please double-check and fix them. For example:

  • 2, line 65: "... sample size, undertake and retention, ... "
  • 8, line 305: "This stand alone service centre conduct one to two professional led group annually..."
  • 9, Table 3, in the row for "Active experimentation" and the column for "Activity": "Roleplay..." should be "Role play..."
  • 14, line 453: "conditiond" should be "conditions"

Response: Thanks for the suggestion. Revisions were made accordingly.

#3) Better word choices would contribute to clearer communication of the authors' ideas. For example:

-- p.1, Abstract, line 9: Suggest to replace "most important purposes" with "most important goals". With the replacement, this programme of work can be linked to the literature on goals and goal pursuit in social psychology. This suggestion may have implications for the underpinning science-- it is up to the authors to decide whether to follow this suggestion.

-- P.9, line 350: Suggest to replace "wider readers" with "a bigger audience".

-- P.9, line 364: Suggest to replace "can be evidenced in multi-site pilots..." with "can be tested in multi-site pilot studies..."

-- p.10, line 381-383: Suggest to replace "... and will continue to be practiced with a significant contribution towards enhancing a culturally specific explanation of active and healthy aging globally" with "... and is expected to continue to be practiced and to contribute to the development of a culturally specific conceptualization of active and healthy aging".  It would be prudent to avoid making definite statements when the evidence from rigorous scientific research is not yet available. 

Response: Thanks for the suggestion. Revisions were made accordingly.